# Telerehabilitation with ARC Intellicare to Cope with Motor and Respiratory Disabilities: Results about the Process, Usability, and Clinical Effect of the “Ricominciare” Pilot Study

**DOI:** 10.3390/s23167238

**Published:** 2023-08-17

**Authors:** Marianna Capecci, Rossella Cima, Filippo A. Barbini, Alice Mantoan, Francesca Sernissi, Stefano Lai, Riccardo Fava, Luca Tagliapietra, Luca Ascari, Roberto N. Izzo, Maria Eleonora Leombruni, Paola Casoli, Margherita Hibel, Maria Gabriella Ceravolo

**Affiliations:** 1Department of Experimental and Clinical Medicine, Polytechnic University of Marche, 60121 Ancona, Italy; cima.rossella@gmail.com (R.C.); f.a.barbini@staff.univpm.it (F.A.B.); robertonicolaizzo@gmail.com (R.N.I.); mariaeleonoraleombruni@gmail.com (M.E.L.); p.casoli@staff.univpm.it (P.C.); margheritahibel@yahoo.it (M.H.); m.g.ceravolo@univpm.it (M.G.C.); 2Henesis Division, Camlin Italy Srl, 43123 Parma, Italy; a.mantoan@henesis.eu (A.M.); f.sernissi@henesis.eu (F.S.); s.lai@henesis.eu (S.L.); riccardo.fava@henesis.eu (R.F.); l.tagliapietra@henesis.eu (L.T.); l.ascari@henesis.eu (L.A.)

**Keywords:** telerehabilitation, wearable sensors, inertial measurement units, artificial intelligence, Parkinson’s disease, COVID-19, disability, dyspnea, fatigue, adherence

## Abstract

Background: “Ricominciare” is a single-center, prospective, pre-/post-intervention pilot study aimed at verifying the feasibility and safety of the ARC Intellicare (ARC) system (an artificial intelligence-powered and inertial motion unit-based mobile platform) in the home rehabilitation of people with disabilities due to respiratory or neurological diseases. Methods. People with Parkinson’s disease (pwPD) or post-COVID-19 condition (COV19) and an indication for exercise or home rehabilitation to optimize motor and respiratory function were enrolled. They underwent training for ARC usage and received an ARC unit to be used independently at home for 4 weeks, for 45 min 5 days/week sessions of respiratory and motor patient-tailored rehabilitation. ARC allows for exercise monitoring thanks to data from five IMU sensors, processed by an AI proprietary library to provide (i) patients with real-time feedback and (ii) therapists with information on patient adherence to the prescribed therapy. Usability (System Usability Scale, SUS), adherence, and adverse events were primary study outcomes. Modified Barthel Index (mBI), Barthel Dyspnea Index (BaDI), 2-Minute Walking Test (2MWT), Brief Fatigue Inventory (BFI), Beck Depression or Anxiety Inventory (BDI, BAI), and quality of life (EQ-5D) were also monitored pre- and post-treatment. Results. A total of 21 out of 23 eligible patients were enrolled and completed the study: 11 COV19 and 10 pwPD. The mean total SUS score was 77/100. The median patients’ adherence to exercise prescriptions was 80%. Clinical outcome measures (BaDI, 2MWT distance, BFI; BAI, BDI, and EQ-5D) improved significantly; no side effects were reported. Conclusion. ARC is usable and safe for home rehabilitation. Preliminary data suggest promising results on the effectiveness in subjects with post-COVID condition or Parkinson’s disease.

## 1. Introduction

The Global Burden of Diseases, Injuries, and Risk Factors Study 2019 estimated that globally, in 2019, 2.41 billion individuals had conditions that would benefit from rehabilitation [1]. This number had increased by 63% from 1990 to 2019. The disease area that contributed most to prevalence was musculoskeletal disorders, followed by neurological disorders [1], at least until SARS-CoV-2-related pandemic disrupted life and good clinical practice worldwide.

Cieza et al. pointed out that one-third of people with a disease or trauma require rehabilitation, so rehabilitation should be an integrated asset in the acute and chronic care pathway, available close to the patient and equally accessible. Instead, rehabilitation has not been prioritized and is under-resourced [1]. Rehabilitation has been shown to be effective in clinical terms but also in reducing health-related costs: it can prevent complications during acute hospitalizations and reduce length of stay [2,3]. By working on people’s function and abilities, rehabilitation helps people to be independent and recover their role in society [4,5]. Advanced, digital technological solutions have become commonplace and are increasingly applied in rehabilitation to optimize care, such as through robotic devices or by reaching people remotely [6,7].

Technologies could support clinical practice, especially considering that rehabilitation is a therapy based on the neurophysiological principles of repetitive, intensive, specific, task-oriented practice. In addition, continuity of rehabilitation care is a key element of effectiveness in preventing the progression of disability for patients with chronic conditions. Among technological solutions, telerehabilitation may be the answer to the needs for effectiveness and efficiency, continuity, and sustainability of care, in the post-acute phase, such as after a stroke [8], and for managing chronic disabilities in both neurological and non-neurological conditions [9,10,11,12]. However, there is no formal structure for the provision of telerehabilitation services, and data exchange can take many forms [13].

Until 2020, telerehabilitation had remained mostly confined to experimental solutions, which found little application in clinical practice. Inadequate laws, the problem of cybersecurity and privacy, heterogeneity of systems, incomplete architectures, a lack of verification of automatic signal evaluation algorithms, and the low technological literacy of patients and physicians are some of the limitations that have prevented the widespread deployment of telerehabilitation in clinical practice. In 2020, the coronavirus infection disease-19 (COVID-19) pandemic [14,15] and the need for social distancing promoted the immediate real-world use of telemedicine and telerehabilitation solutions. The aim was mainly to reach (i) all those at home with chronic conditions and (ii) the multitude of people recovering from COVID-19 itself [16].

An archetype of progressive and disabling chronic disease is Parkinson’s disease (PD). PD is one of the two most common progressive neurodegenerative disorders, along with Alzheimer’s disease, but disability and death due to PD are increasing faster than for any other neurological disorder [17]. It has become a major global concern for the WHO because the current prevalence (i.e., 1% of the population over the age of 60 [18]) has doubled in the past 25 years, and this increase is becoming even more rapid. In 2019, PD resulted in 5.8 million disability-adjusted life years, an increase of 81% since 2000 [19]. PD is associated with motor symptoms (slow movement, tremor, rigidity, and imbalance) and other complications, including respiratory changes, cognitive impairment, mental health and sleep disorders, pain, and sensory disturbances. Scientific evidence shows that levodopa is the most effective medication for improving symptoms, functioning, and quality of life. In addition, physical activity and rehabilitation are the necessary therapeutic complement, from the early to the terminal phase of PD, to optimize functioning and quality of life. However, care is not accessible, available, or affordable everywhere and at all times, notwithstanding continuity of multidisciplinary care being a key to therapeutic success in the management of PD [20,21,22,23]. During the COVID-19 pandemic, changes in medical care, activities of daily living, social support and physical activity led to worsening motor and non-motor symptoms in people with PD (pwPD) [24]. An increase in anxiety, chronic stress, and depression was correlated with reduced opportunity for regular physical activity and difficulty accessing physiotherapy [25,26,27]. Approximately 80% of pwPD discontinued physiotherapy treatments and up to 60% reported worsening symptoms [28,29].

Globally, from the beginning of the pandemic until April 2023, more than 760 million confirmed cases of COVID-19 have been reported to WHO, including 6.9 million deaths (https://COVID19.who.int/ accessed on 3 August 2023). Eleven percent of patients with laboratory-verified disease were admitted to intensive care units (ICU), (twice as many as for seasonal influenza [30]) for severe acute respiratory syndrome. Due to impaired neuromuscular function or prolonged periods of immobility, such a syndrome can lead to post-infectious impairment of physical function, which can persist beyond 12 months in more than 20% of cases [31].

Six months after SARS-CoV-2 infection, the objective measure of physical functioning showed below-normal performance with reduced respiratory muscle strength. After 12 months, patients still complained of residual symptoms. A syndrome characterized by fatigue, shortness of breath, and cognitive dysfunction impacting daily functioning and occupational activities has been described as a separate nosography condition after COVID-19. This syndrome may be a new onset condition 2–3 months after infection, may appear after initial recovery from an acute episode of COVID-19, or persist from the initial illness. Subjects with advanced age, those who had been admitted to the intensive care unit in the acute phase, and those who complained of multiple symptoms at the onset of COVID-19 are more likely to suffer from long-term symptoms, negatively impacting physical and mental well-being [24,31]. These patients experienced a lack of clinical support and contradictory advice regarding rehabilitation [32]. During the pandemic, two million patients, in Europe, did not have access to rehabilitation services, due to the stop of admissions, early discharge, and a reduction in activities [33]. After the pandemic, recovery of usual healthcare practice was difficult due to increased demand for respiratory, motor, and non-motor rehabilitation. However, pre–post studies suggested that 6–8 weeks of ambulatory resistance and strength training could effectively relieve respiratory symptoms and increase endurance in adults with a post-COVID condition [34,35]. Training for a return to physical function after infection needs to be better coordinated, and guidelines for healthcare providers are needed to prevent patients from receiving conflicting advice [36]. However, to achieve these goals, we need more evidence of effectiveness and tools that improve accessibility to services.

In clinical trials conducted in the hospital setting, telerehabilitation has been found effective in shortening hospital stays, facilitating home discharges, and providing education and support to patients and caregivers [11,37,38,39]. On the contrary, in the outpatient setting, telemedicine has complemented or replaced face-to-face encounters in acute and chronic neurological, cardiac, and musculoskeletal conditions commonly treated by physiatrists [13]. Actually, we can intend telerehabilitation not only as a remote system to provide rehabilitation but also as the delivery of healthcare services via information and communication technologies.

Several systematic reviews conclude that telerehabilitation is effective for patients with neurological and musculoskeletal diseases in recovering motor function [40,41,42,43]. Some studies suggest that it can also reduce healthcare costs, increase treatment adherence, improve mental function and quality of life, and be delivered satisfactorily for remote patients [43,44,45,46]. Most studies on telerehabilitation are concerned with the outcomes of synchronous, real-time rehabilitation, although there is some evidence that asynchronous telemedicine can be effective [47].

More in-depth studies are therefore needed to answer questions about the feasibility, safety, and efficacy of telerehabilitation modalities in subgroups of patient populations and settings, such as those frail people who are at risk of disability progression due to lack of rehabilitation care or for motor disorders, falls, and respiratory symptoms [5,48].

For this scope, we selected, as case studies, subjects with post-COVID-19 condition and PD as widespread archetypes of multisystem chronically disabling acquired or progressive disease, respectively. Moreover, we used the ARC Intellicare (ARC) system to provide telerehabilitation. ARC is an artificial intelligence (AI)-powered and inertial motion unit (IMU)-based mobile platform enabling personalized rehabilitation for motor and respiratory recovery for patients requiring continuous home rehabilitation. It has been conceived and developed to target post-stroke patients (MAGIC, grant agreement n. 687228), and then optimized to address the above-mentioned emerging needs coming from the COVID-19 pandemic (POR FESR 2014-2020-Asse I COVID-19 grant), following recommendations for people’s recovery after COVID-19 [33,36].

### Study Design and Objectives

‘Ricominciare’ is a pilot single-center, not controlled, prospective, pre–post-intervention study aimed at verifying the feasibility and safety of the ARC platform intended for the home rehabilitation of people suffering from mild to moderate disabilities due to respiratory or neurological conditions related to COVID-19 or PD.

In particular, the primary objectives of the study were to test the feasibility of integrating the ARC Intellicare solution into the care pathway for COVID-19 survivors and pwPD in terms of:Adherence to the home rehabilitation program;Safety of rehabilitation therapy.

Secondary objectives of the study were to investigate:The usability and acceptability of the intervention;The process to provide the new care pathway;Clinical effectiveness: in fact, the participants will undergo pre–post-intervention monitoring of disability in basal activity of daily living (ADL), respiratory outcomes, endurance and fatigue, mood, and quality of life.

## 2. Materials and Methods

### 2.1. ARC Intellicare

ARC is a telerehabilitation solution based on the use of multiple wearable sensors, a mobile device, and algorithms of artificial intelligence (patent pending). ARC consists of a set of 5 inertial sensors (MetaMotionR+ from MbientLab, San Francisco, CA, USA) inserted in slap supports, a tablet with a dedicated application (App), and a charging station (Figure 1).

The mobile App comes with a Home version, with specific features for the patient, and a Clinical version, designed instead for the healthcare professional. The device allows rehabilitation professionals to prescribe exercises from the available library according to specific therapeutic needs and/or according to the rehabilitation protocols adopted by the center, and to monitor patients’ performances and progresses remotely. With the Home version, ARC guides patients in the unsupervised execution of the rehabilitation project through simple instructions, video tutorials, and the automatic counting of the correct number of repetitions performed. The counting of the number of exercise repetitions correctly performed is the output of the developed AI algorithm, which is the innovative core of this device. It runs real-time on the ARC backend deployed in a dedicated cloud server. Input of this algorithm are the tri-axial accelerations and angular velocities. These data are streamed simultaneously via Bluetooth to the tablet from 3 out of 5 IMU sensors worn by the patients during the exercise session. The sensors involved in the monitoring are selected based on the exercise type and the target body area. For upper body exercises and for those mainly involving upper arm movements, the upper limb–trunk sensor configuration is used (i.e., ULT, two on both wrists and one back neck sensors). For all exercises requiring movement mainly on the lower limbs, instead, a lower limb–trunk sensor configuration is used (LLT). The positioning and orientation of the sensors on the body, with their relative axes, are shown in Figure 2.

Accelerations and angular velocities coming from the three pre-defined inertial sensors are buffered in a temporally ordered manner and normalized. A neural network is used to recognize which exercise is being performed and to segment the data into single repetitions. When a repetition is identified by the algorithm, it is considered correctly performed, and real-time feedback is provided to the patient through the App user interface. At the end of each exercise execution, a counting unit outputs the total number of repetitions performed, which is then used to compute patient adherence to the prescribed therapy (refer to Section 2.4). This information, together with other feedback collected directly from the patient on his/her health status during and at the end of each session (patient reported outcomes), is collected and displayed on a results dashboard in the Clinical version of the ARC App.

The accuracy and reliability of the commercial IMU sensor part of ARC Intellicare, i.e., MetaMotionR+, was previously assessed in the literature for both sensor fusion embedded algorithm [49] and raw data (i.e., accelerations and angular velocities) [50]. The AI algorithm was validated on a group of 30 healthy subjects and for all the 41 motor exercises available in the ARC exercise library. The validation consisted of the comparison between the number of exercise repetitions identified by an expert operator (ground truth), N_operator_, and the number of exercise repetitions counted by the algorithm for the same exercise execution (model predictions), N_ARC_. Assuming a normal distribution, a paired t-test between the couples N_operator_ and N_ARC_ (one per exercise) was used to verify the null hypothesis, i.e., no statistical difference between the assessment made by an expert operator and the AI algorithm. In 35 of 41 exercises (86%), no significant difference in relative repetition count between the ground truth and the model predictions (*p* ≥ alpha = 0.05), was obtained, confirming the functional performance of the device on 35 different exercises (validation data are available upon request). Only exercises for which validation was successful have been considered in this study.

The novelty of the ARC device with respect to other commercial solutions is the use of a validated AI algorithm able to recognize the high number of exercises included in the library, varying in typology and complexity level. In the ARC exercise library, indeed, different types of rehabilitation exercises (e.g., for mobility, coordination, balance, core stability, strengthening, etc.) are available, and all of them are already widely used in clinical practice [51]. These exercises are characterized by a periodic movement, which can be composed of several phases and can comprehend the use of different limbs. Five levels of difficulty have been defined based on exercise pattern (i.e., bilateral symmetric—BS, bilateral alternated—BA, mono-lateral—ML, static—ST), exercise starting position (comfortable or uncomfortable), and the number of limbs and joints involved.

### 2.2. Subjects

People (men and women > 18 years) following COVID-19 (COV19) or diagnosed with PD were enrolled between 26 March 2021 and 30 May 2021 if received by a physiatrist an indication for rehabilitation to cope with motor or respiratory disorders in order to optimize the independence in activity of daily living (ADL). Inclusion criteria were (i) mild–moderate dyspnea (Barthel dyspnea Index ≤ 95) [52] or (ii) Walking Handicap Scale (WHS) [53] ≤5; and (iii) signed informed consent. The study exclusion criteria were (1) at the enrollment, presence of fever (TC ≥ 37 °C), cough, cold, sore throat, diarrhea, or pneumonia signs and diagnosis of moderate to severe cognitive impairment; (2) formal rehabilitation performed in the last month; (3) pre-existing disability related to dementia, epilepsy, seizures, and a history of severe dizziness and falls; (4) severe non-stabilized comorbidities, such as oncological diseases in the active phase, New York Heart Association (NYHA) Functional Classification stage IV congestive heart failure [54], or severe respiratory failure requiring cough and breath support; (5); Rankin mod. Score ≥ 4 [55], i.e., moderate or severe dependence in activities of daily living for any medical reason (but we accepted an inconstant need for help to use technology or a supervision to prevent falls during motor training); (6) for women of potential childbearing, not using suitable valid methods of contraception; (7) pregnancy.

### 2.3. Intervention Protocol

Upon obtaining informed consent, each participant underwent clinical–functional evaluation, usability tests, and training in ARC use, receiving an ARC unit, to be used independently at home in the next 4 weeks.

Regarding the training, at the enrollment, each subject received from the site investigator a 30 min explanation on the use of both software (SW) and hardware (HW) components. During this training, the patient also performed some of the exercises in the rehabilitation protocol that had been explained to him/her. Once completed, each subject carried out a usability test, in which he/she was asked to perform 15 tasks (6 related to HW and 9 to SW components). For each task, the patient was asked to indicate if he/she needed support from the investigator and the degree of difficulty encountered in carrying it out on a scale from 0 to 10 (0 = no difficulty and 10 = impossible to perform). Each subject then completed the System Usability Scale (SUS) [56]. After enrollment, each subject underwent the clinical assessment and received an ARC unit to perform 45 min exercise sessions at home, 5 days/week for 4 weeks and an at least 30 min/week video call with the investigator using the ARC app. The use of ARC during unsupervised sessions allows for exercise and patient adherence monitoring.

### 2.4. Study Endpoints and Outcome Measures

Primary study endpoints and outcome measures (Table 1) were:
To reach an effective adherence to the home rehabilitation program (at least 80%), measured as the rate of performed/prescribed sessions [57];Safety of rehabilitation therapy, based on the number/type of adverse events [58].

Considering motor and respiratory data separately, we proposed two different ways to calculate adherence to home-based exercises prescribed with ARC:    1.Adherence-Days = Total number of days the patient accessed the platform for training versus the total number of days the exercises were prescribed (1);
(1)$$\sum_{d=first}^{last}Adh_{thresholded}\left(d\right)$$
where *Adh_thresholded_* is computed using a threshold on the number of daily exercise executions (*N_executions_*), considering together all exercises prescribed for that day. In particular:*Adh_thresholded_* (*d*) = 0, when the patient never tried to access into ARC device to perform one of the exercises prescribed for day *d*;*Adh_thresholded_* (*d*) = 1, when *N_executions_* (*d*) ≥ 1.

    2.Adherence-Repetitions = total number of repetitions performed versus total number of repetitions prescribed, considering all exercises included in the rehabilitation plan (*e*, from 1 to *n*, where n is the total number of exercises prescribed to a subject) and all days (*d*) of treatment (from *d = first*, i.e. first day of treatment to *d = last*, i.e. last day for which an individual rehabilitation program was prescribed) (2).


(2)
$$\sum_{d=first}^{last}\sum_{e=1}^{n}Reps{\left(e\left(d\right)\right)}_{performed}/\sum_{d=first}^{last}\sum_{e=1}^{n}Reps{\left(e\left(d\right)\right)}_{prescribed}$$


In the first analysis, we evaluated the number of days each patient used ARC compared to the total days they had a prescription, despite the number and type of exercises prescribed by the rehabilitation professional, and without considering if the patient completed them or not.

In the second analysis, we evaluated the number of repetitions that each patient performed with ARC compared to the total repetitions assigned by the rehabilitation professional. All sets/exercises prescribed are considered in this analysis. Both adherence calculations were expressed as a percentage.

Secondary endpoints and outcome measures (Table 2) were:usability and acceptability of the intervention studied through the System Usability Scale (SUS) [56] and a semi-structured ad hoc-prepared questionnaire;The process to provide the new care pathway, measured by the percentage of subjects resulting eligible to the study.

In addition, pre- and post-treatment clinical outcomes were monitored as follows:Disability: modified Barthel Index (mBI) [59];Respiratory outcomes: Barthel Dyspnea Index (BaDI) [52]; and Borg Scale [60] during Two-Meter Walking Test (2MWT) [61];Motor outcomes: 2MWT [61];Fatigue: Brief Fatigue Inventory (BFI) [62];Mood and anxiety: Beck Depression or Anxiety Inventory (BDI, BAI) [63];Quality of Life: Euro-Quality of life Questionnaire self-assessment-5 Dimension (EQ-5D) and EQ-5D-Visual Analogic Scale (EQ-5D-VAS) [64].

In Table 1 and Table 2, we also reported the cut-off scores of the outcome measures of the different end-points based on the availability of the minimum clinically important difference (MCID) score in the literature.

The assessment was performed at the enrollment (T0) and at the end of the 30-day intervention period (T1), when monitoring of primary and secondary outcome measures on adherence, acceptability, and safety was completed.

At the enrollment, the following data were recorded: age, gender, years of education, body mass index (BMI), Media and Technology Usage and Attitudes Scale (MTUAS) [65], Montreal Cognitive Assessment (MoCA) [66] and Walking Handicap Scale (WHS) [67].

All the secondary endpoint outcome measures were assessed at the University Hospital outpatient facilities, at each assessment time point, by blinded clinicians who used standardized questionnaires sheets and devices (finger pulse oximeter, OXY Watch ChoiceMMed, Pikdare SpA; Hensych Blood Pressure Cuff Kit with Manual Sphygmomanometer and Stethoscope) independent from ARC. Participant selection and enrollment were performed by physiatrists MGC and MC while the blinded assessment of study outcomes was performed by physiatrists FAB and MEL. The physiatrist RC and the physiotherapists MH, PC, and RI built the exercise library. RC and PC supervised patients during the rehabilitation period, selecting and updating the training protocol, if needed, during the weekly synchronized training sessions. All raters involved in the study underwent a preliminary course to harmonize methods and increase inter-rater reliability to a Cronbach alpha 0.8.

**Table 2 sensors-23-07238-t002:** Secondary objectives and related outcome measures and endpoints.

Secondary Objectives	Outcome Measures	Endpoint	Ref.
Process description of the care pathway Acceptability of intervention (tech)	Percentage of subjects resulting eligible to the study Percentage of subjects who accept to undergo telerehabilitation	n.a. n.a.	n.a. n.a.
Usability of the intervention device (tech)	System Usability Scale (SUS)	>70%	[56]
Clinical	modified Barthel Index (mBI) Barthel Dyspnea Index (BaDI) Two-Minute Walking Test (2MWT) Borg Dyspnea Scale Brief Fatigue Inventory (BFI), Beck Depression or Anxiety Inventory (BDI, BAI), Quality of Life (Euro-Quality of Life Questionnaire self-assessment-5 Dimension (EQ-5D); EQ-5D-Visual Analogic Scale (EQ-5D-VAS))	−2 points −9 points +(4–11) m +1 point n.a. n.a. n.a.	[59,68] [52,69] [61,70,71] [60,71,72] [62] [63] [64]

Legend: n.a. = not applicable because not logical or because the data are not in the literature.

### 2.5. Ethical Procedures

The study was performed according to the Declaration of Helsinki and approved by the Local Institutional Committee (protocol number: CERM1781). All participants signed informed consent forms prior to participation in the study. Trial registration: the trial ClinicalTrials.gov Identifier is NCT05074771.

### 2.6. Statistical Analysis

At least 20 subjects (10 subjects per group: i.e., COV19 group and pwPD) were expected to be enrolled in this exploratory study. The sample size was calculated by the confidence interval method (IC95%), using the Clopper–Pearson exact method, and defining a 95% confidence level. A sample size of 10 subjects per group was needed to estimate an adherence (calculated as the percentage of day accesses) of at least 80%, with an interval width of 0.8 (IC95% 0.44–0.97). Such a sample of 20 subjects was deemed appropriate to achieve results characterized by an error of the estimate compatible with the exploratory nature of the study.

The distribution of variables in the whole sample as well as in the 2 sub-groups (COV19 and pwPD) was determined using mean and standard deviation, and number/percentage for categorical variables.

Pre–post-treatment within-group changes were checked by using the Wilcoxon rank test for the whole sample as well as for each study group. The Mann–Whitney U test with Z analysis was applied for comparing data across the 2 groups, after verifying the normal distribution through the skewness and kurtosis tests.

Finally, to control the influence exerted by a patient’s personal and clinical characteristics on the treatment effect, firstly we calculated a change index delta = ([t1 score − t0 score]/t0 score) × 100 for each clinical outcome, whereas we applied the Spearman rank correlation analysis to study the effect of age, MTUAS, MoCA, and years of education on delta score.

Statistical significance was set at the 0.05 level. The analysis was performed using Statview Statistics, version 5.0.

## 3. Results

### 3.1. Population and Process Description

Thirty-six (46%) out of seventy-eight (38 COV19 and 40 pwPD) subjects undergoing a physiatrist visit during the study period met the eligibility criteria: eighteen (50%) suffered from post-COVID-19 condition and eighteen (50%) from pwPD. Among these 36 subjects, 13 (36%)—6 COV19 and 7 pwPD, respectively—declined to sign informed consent because of lack of confidence with technology and no caregivers to support the home rehabilitation (1 COV19, 4 pwPD), reasons that were not clinical (2 COV19), preference for in-presence outpatient rehabilitation (2 COV19, 3 pwPD), or no time to perform rehabilitation (2 COV19).

Of the 23 subjects receiving training sessions (12 COV19 and 11 pwPD), 2 were defined as unsuitable due to limitations in the use of technology because of limitations in using the technology and a lack of caregivers who could help them use it; therefore, 21 (mean age 61 ± 10 years [range: 29–72], 8 women) were enrolled, and all completed the study: 11 were COV19 (mean age 57 ± 13 years [range: 29–72], 5 women) and 10 were pwPD (mean age 65 ± 4 years [range: 59–70], 3 women). Figure 3 shows the flow diagram of the Ricominciare study.

Table 3 shows the detailed demographic and clinical data of the enrolled subjects.

The MTUAS score is slightly related to age (Spearman corr: Z = −2.0; *p* = 0.048) and highly with MoCA score (Z = 2.8; *p* = 0.005).

### 3.2. Usability and Acceptability

None of the COV19 subjects needed support for using the device after training; 4 out of the 10 pwPD needed a caregiver during home sessions. In the 1-month period of ARC use at home, forty-two technical support requests were raised from the twenty-one patients that completed it (i.e., an average of two technical support tickets per patient). No issues prevented the study from being completed successfully. Indeed, those related to hardware components, leading to the replacement of a sensor or charging station, occurred in 3 out of 420 ARC usage sessions (i.e., 0.7% of the overall system usage). These issues caused training discontinuation for 2 days on average (corresponding to the time needed for component substitution). All other issues were generally solved by restarting the device/app, and thus not hindering patients from completing the prescribed rehabilitation program. Twenty-four issues (5.7%) were related to the management of the Bluetooth connection between the tablet and sensors. Four issues (1%) were caused by 3G/4G or WiFi network connectivity, while four issues (1%) were linked to the real-time feedback provided by the device during exercise execution. Finally, the remaining seven tickets (1.7%) were not triggered by actual technical problems and were solved by clarifying to the user how to properly use the system and where to find the requested functionality.

At the baseline, the mean total SUS score was 77/100 (±14) (COV19: 80 (±12) and pwPD: 74 (±15)). After 30 days of treatment, the SUS remained stable at 78/100 (±14) (COV19: 75 (±16) and pwPD: 80 (±13)) (*p* = 0.91). No statistically significant differences were found between groups at the baseline (*p* = 0.15) and follow-up (*p* = 0.50).

### 3.3. Adherence

Table 4 shows mean values and related standard deviation, both based on total prescription days—i.e., Adherence-Days—and based on the total number of repetitions assigned—i.e., Adherence-Repetitions—considering the overall study population (total) and the two groups (COV19 and pwPD) individually for motor and respiratory data, respectively. On average, the whole sample of subjects achieved 80% adherence calculated on days and 77% on repetitions. In particular, sixteen subjects (76%) reached the endpoint (80% of Adherence-Days). Measuring Adherence-Repetitions, 15 (76%) reached the endpoint of 70%.

No differences emerged between the disease groups. Adherence remained stable, on average, from the first to the last week; however, at first, pwPD exhibited low Adherence-Days (i.e., 70%), which increased significantly to a stable average of 80% the following week (Z = −3.1; *p* = 0.002).

### 3.4. Clinical Data Evolution

Table 5 shows the results of descriptive and pre–post-treatment comparative statistics. In the whole sample, the dyspnea assessed by the BaDI improved significantly (Z = −3.0; *p* = 0.003) (Figure 4), as well as fatigue, according to the BFI (Z = 3.5; *p* = 0.005) (Figure 5). The 2MWT distance (meters) improved significantly (Z = −3.2; *p* = 0.001) (Figure 6) without statistically significant changes in heart rate or SpO^2^, which fell within normal values. Mood state (BDI) (Z = 3.1; *p* = 0.002) and anxiety (BAI) (Z = 2.5; *p* = 0.01) improved significantly, as well as quality of life (EQ-5D: Z = 2.6; *p* = 0.0084; EQ-5D VAS percentage: Z = −2.7; *p* = 0.007) (Figure 7).

Stratifying by pathology, at the baseline, no significant differences between groups emerged. COV19 improved in all outcome measures, while pwPD improved in all measures, except for 2MWT, whose performance improved without reaching statistical significance.

No correlation between demographic (age, education) or clinical data (MoCA, modified Rankin disability score, MTUAS, SUS) with respect to clinical effects (delta score of BaDI, mBI, BFI, 2MWT, and EQ.5D) was found using the Spearman rank test (*p* > 0.05); nor between adherence measures and clinical effects.

No adverse events were reported except for fatigue in 10% of pwPD.

## 4. Discussion

The analysis of the flow of subjects through the study showed that sixty-seven out of seventy-eight (83%) examined people with COVID-19 or pwPD outcomes needed rehabilitation. Among them, twenty-nine (37%) were already undergoing rehabilitation, while thirty-six (46%) were eligible for the study. However, seven subjects (9% of the whole sample, most pwPD) showed difficulty in using or accepting the technology, while five (6%) preferred to undergo outpatient in-person rehabilitation and did not sign the informed consent. Firstly, those numbers reflect the high frequency of outcomes needing rehabilitation in both post-COVID-19 condition (80% of participants, irrespective of the results of the pulmonary function tests according to Ceravolo et al., 2021) [73] and PD [19,20,21,22,23].

Secondarily, the observed prevalence of people (46%) who were not undergoing rehabilitation despite needing it may reflect a persistent difficulty that had been observed in 2020, during the first phase of the COVID-19 pandemic, when up to 80 percent of pwPD had to discontinue or were unable to access physiotherapy treatments due to social distancing, fear of contagion, or hospital difficulties [28,29].

The last result that emerges from the study enrollment phase is the level of difficulty in using or accepting the technology and remote rehabilitation: this was manifest in less than 10% of the whole sample and less than 5% among people receiving training.

For effective use, technologies in healthcare require rigorous validation to prove their acceptability and usability in addition to clinical benefits [74]. The “acceptability” of technology is a tradeoff among all those factors that influence the adoption of new technologies [75]. Two key dimensions may characterize the users’ acceptability: the “perceived ease of use” and the “perceived usefulness” according to the Technology Acceptance Model (TAM) [76].

Moreover, additional external factors may influence the user acceptance of technology, including demographic and clinical characteristics (i.e., age, education, and disability level), technological usage and attitude, and social or cultural influences, usability, availability, assurance of privacy and security [77,78]. Assurance of privacy seems a critical concern for older adults, followed by the functionality of the system [79]. Moreover costs, usability (both as ease of use and suitability for daily use), stigma and technical support are critical barriers to digital health adoption [79,80]. Considering the usability issue, we selected the SUS score as an important outcome of the study: poor usability is one of the main causes of technological system abandonment [75,76] and was shown to influence people’s acceptance of digital solutions and adherence to the treatment [79].

In our study, the level of usability of the ARC solution, as measured by the SUS [56], is in line with that reported by Rossetto and colleagues (2023) [74], who studied a telerehabilitation system in chronic disabilities, including chronic obstructive pulmonary disease and PD. On average, the SUS score was above 70/100, recognized as an appropriate cut-off of good acceptance [56]. In our sample, pwPD, who were older and more disabled than COV19, presented slightly lower acceptability and usability of telerehabilitation at the time of enrollment, but after the study, the level of reported ARC usability improved slightly. It is possible that ARC met some of the requirements that several authors [58,75,80,81,82,83,84,85,86] believe are important for improving usability and adherence, such as the establishment of a good therapeutic alliance with patients through initial training, the ability to choose and adapt the physiotherapy program at any time tailored to the patient, and to reach goals important to the patient. These characteristics may also be useful in the case of COVID-19, which showed a good baseline value that was slightly lowered. It is possible that some technical problems encountered during rehabilitation may affect the SUS value at follow-up.

The quality and efficacy of healthcare depends on patient adherence to recommended treatment regimens. Poor adherence to treatment may lead to worse outcomes across many healthcare disciplines including physiotherapy [87,88].

The extent of adherence to physiotherapy outpatient or home-based treatment varies from 30% [89,90] to 70% [81]. This high variability depends firstly on different and not validated measures proposed to quantify it, both subjective and objective [57,58], and secondarily on a range of contributing factors [58,91], such as (i) patient-related factors: clinical (i.e., disease severity, poor self-efficacy), demographic (i.e., gender, age), cognitive/psychological (i.e., dementia, depression, stigma), motivation; (ii) organizational factors (i.e., inability to fit exercises into daily life) [92,93]; (iii) physiotherapy program characteristics: absence of supervision during learning sessions, program design, complexity, low flexibility, burden of exercising [92,93,94]; and (iv) care providers’ style: a lack of monitoring or feedback [90,93,94].

Poor adherence has implications for subjects’ health and well-being, as well as treatment cost and effectiveness [65,87]. The clinical effectiveness of training depends on repetition as well as salience, specificity, difficulty, shaping, motivation, and feedback [8,95,96].

In our study, adherence is the primary outcome, and was set at a high score of 80% to obtain the theoretical maximum effect based on daily exercise [57,58]. ARC Intellicare allows for the monitoring of both daily access to the platform and the number of repetitions of the gesture. By monitoring the number of repetitions and—through the developed algorithm—the correct execution of the gesture, it allows us to give patients real-time feedback on goals and performance, which are crucial for motor learning [95,96].

Our study’s small sample performed very consistently and with high compliance in both parameters monitored by ARC Intellicare. The measure of repetitions was related to the trend in adherence measured by daily accesses to the platform, although it was lower in percentage value. Adherence measured by repetitions might be a more accurate variable for measuring learning progress: in any case, no differences emerged between the two variables with respect to effectiveness.

A recent scoping review by Yen and colleagues [97] found that the effectiveness of telerehabilitation in promoting independence in activities of daily living (ADLs) is supported by the integration of sensing tools coupled with processing algorithms, which enable remote monitoring and quantifiable measurements. Versatile smart sensors can generate clinically relevant data and recognize user gestures in real time. Machine learning algorithms should be included to promote the learning process. Although remote rehabilitation was conceived with the intention of reducing the healthcare burden and reaching the most distant people, it is still important to ensure—during telerehabilitation as well as in presence—discussion with rehabilitation professionals and adaptability/progression of training in the context of selecting new goals in parallel with learning [97]. Therefore, it is useful to promote mixed models of synchronous and asynchronous rehabilitation and to allow for video consultation with the practitioner. Finally, it might be useful to provide the user with a digital tool literacy program.

ARC allowed for a synchronous session of rehabilitation, that in the study was provided weekly to verify progression and acquire patients’ feedback. Treatment efficacy was demonstrated by a homogeneous and statistically significant improvement, which was observed at follow-up in all selected clinical outcome measures except mBI. The mBI score, although improved, did not reach statistical significance, particularly in the COV19 group, where it already started from very high values (99 on average) at baseline. The chronic condition, typical of both enrolled patient groups, determines the mBI results in a measure that is not sensitive to change by a ceiling effect.

This study has two main limitations. The first is the small sample size, justified by the pilot scope of the study, and the latter is the absence of a control group.

To reduce the bias due to the uncontrolled design and to increase the reliability of interpretation, we measured the clinical outcome through internationally validated scales (Barthel Dyspnea Index [52], Brief Fatigue Inventory [62], etc.) or tests (i.e., 2MWT) [61], and considered the minimum clinically important difference (MCID) value, when available, as a comparison. At the end of the study protocol, wearable sensor-assisted home telerehabilitation, performed through a patient-tailored schedule and with asynchronous sessions at 80%, was effective in increasing patients’ scores beyond the MCID of Barthel’s Dyspnea Index [69], modified Barthel’s Index [68], 2MWT [61,71] and Borg Scale [71,72]. No MCID values are available for the other outcome measures. The pwPD did not achieve a statistically significant improvement in the 2MWT, despite having on average walked 8.1 m more at the end of treatment and reporting less fatigue and a better quality of life. However, walking tests may be widely dependent on the effect of medication rather than cardio-fitness level in moderate–advanced pwPD, such as those enrolled in our study (Hoehn and Yahr score = 3, which means moderate–advanced PD phase) [97]. On the other hand, the BFI score improved significantly in COV19 subjects as well as in pwPD. In pwPD, fatigue could predict the progression of motor dysfunction severity over a longitudinal duration in subjects with disease progression, having a decline in physical and mental fatigue [98].

## 5. Conclusions

Telerehabilitation through the ARC Intellicare system is usable and safe for home rehabilitation in subjects with chronic motor and respiratory disabilities. Preliminary data suggest promising results on the effectiveness in subjects with post-COVID-19 condition or PD. Devices as ARC Intellicare, based on wearable sensors and ML processing algorithms, allow for real-time monitoring of several aspects of adherence: both daily adherence and repetition based on exercise recognition. The results obtained should be confirmed through a larger-sample randomized controlled trial.

Finally, these preliminary results suggest that after rigorous testing, ARC could be useful in the treatment of several conditions causing disability that require motor and respiratory rehabilitation that can be performed remotely.

## Figures and Tables

**Figure 1 sensors-23-07238-f001:**
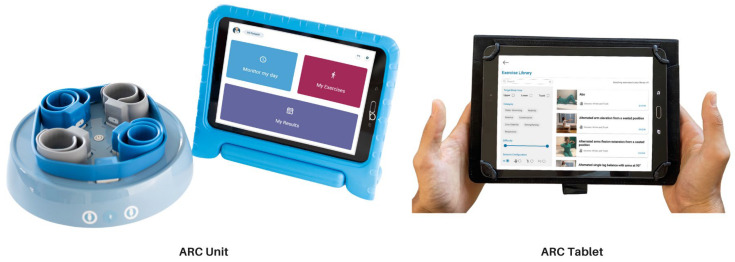
ARC Intellicare device for patient use (Home Unit): a portable and lightweight charging station hosts 5 wearable inertial sensors and their biocompatible supports (2 blue-colored, longer for ankle placement; 2 grey-colored, shorter for wrist positioning; and a longer and more flexible support to be placed on back-side of the neck). The tablet, provided with an anti-shock cover, comes with a preinstalled application with a user-friendly interface and functionalities dedicated to the patient (Home version).

**Figure 2 sensors-23-07238-f002:**
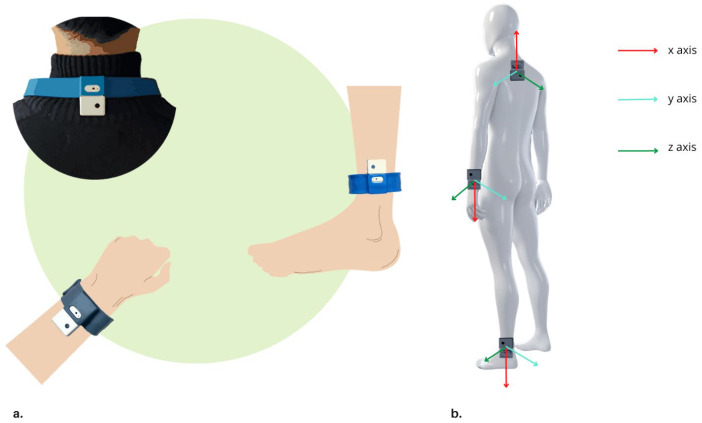
Body positioning of the IMU sensors: back of the neck, on the upper wrists, and above the lateral malleoli (**a**). Sensors are enclosed in white cases and fixed to supports specifically designed to facilitate their wearability. Corresponding reference systems are shown on the right (**b**).

**Figure 3 sensors-23-07238-f003:**
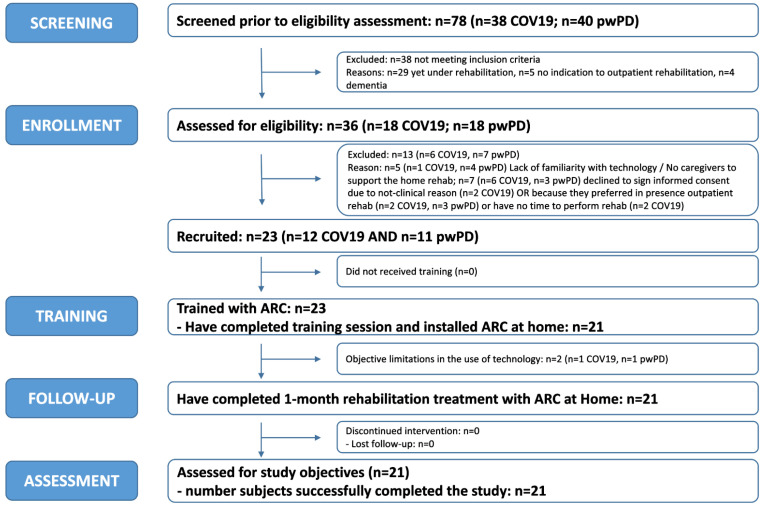
Flow diagram of the Ricominciare study.

**Figure 4 sensors-23-07238-f004:**
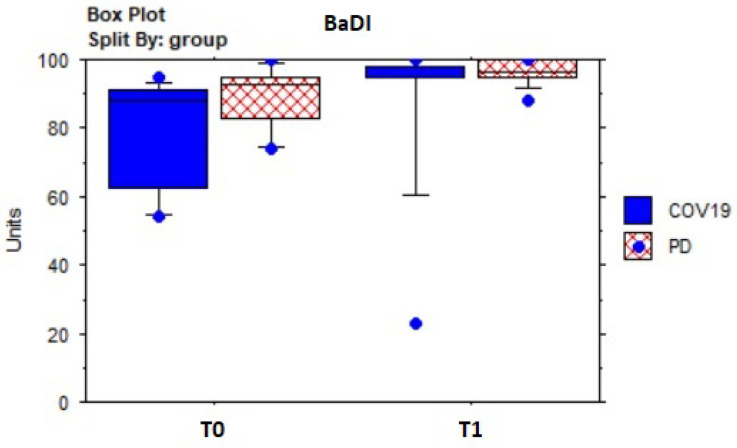
Box plot of Barthel Dyspnea Index (BaDI) score evolution over time by groups.

**Figure 5 sensors-23-07238-f005:**
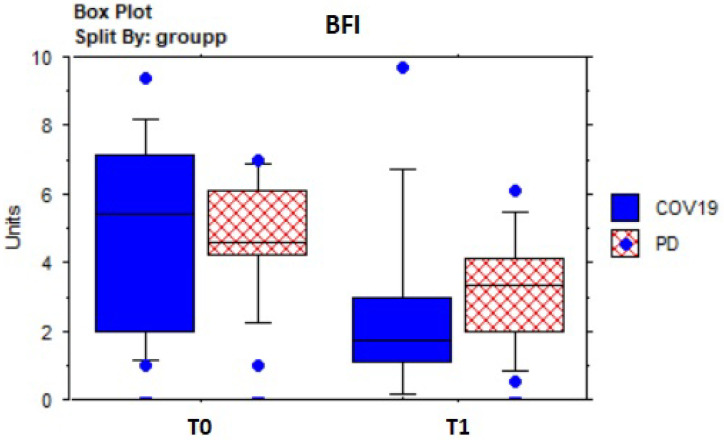
Box plot of Brief Fatigue Inventory (BFI) score evolution over time by groups.

**Figure 6 sensors-23-07238-f006:**
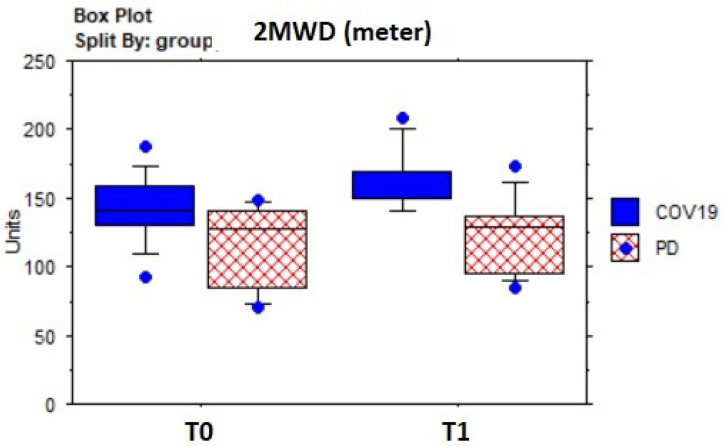
Box plot of 2MWT distance (2MWD) (meters) evolution over time by groups.

**Figure 7 sensors-23-07238-f007:**
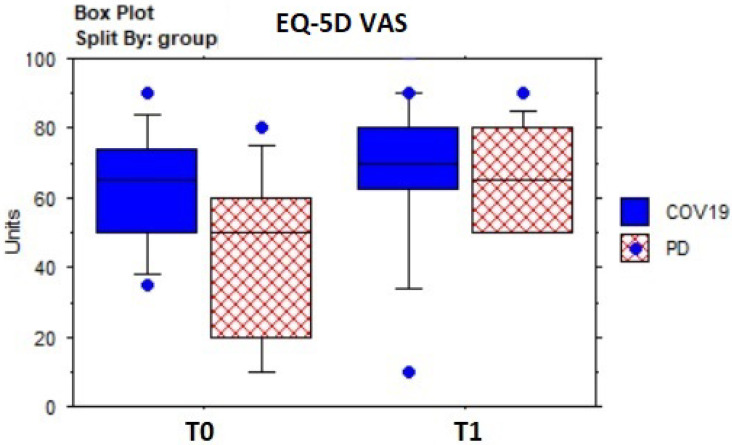
Box plot of EQ-5D VAS score evolution over time by groups.

**Table 1 sensors-23-07238-t001:** Primary objectives and related outcome measures and endpoints.

Primary Objectives	Outcome Measures	Endpoint	Ref.
Adherence to home rehabilitation program	Exercise adherence (days) Exercise adherence (repetitions)	80% 70%	[57,58]
Safety	Number of unanticipated serious device-related adverse effects (USADE), calculated on the total number of adverse events (AE) reported.	0	n.a.

Legend: n.a. = not applicable because not logical or because the data are not in the literature.

**Table 3 sensors-23-07238-t003:** Demographic and clinical data of enrolled subjects.

	Total		COV19		pwPD		COV19 vs. pwPD COMPARISON	
	Mean	Std. Dev.	Mean	Std. Dev.	Mean	Std. Dev.	Z Value	*p* Value
Age	61.1	10.5	57.3	13.2	65.4	3.7	−1.7	0.09
Gender	7 F/14 M		5 F/6 M		2 F/8 M			
Education (Years)	12.5	4.2	13.9	3.8	11	4.2	−1.5	0.12
Hospitalization length of stay (days)	13.2	22.3	26.3	25.8	n.a.	n.a.		
Disease duration (years)					9	2.3		
DAYS FROM COVID19			117	82				
RANKIN disability score	1.4	1.0	1	0	1.8	1.3	−1.2	0.24
WHS (T0)	5.3	1.0	5.5	0.52	5.2	1.3	−1.9	0.62
MTUAS	213	69.6	235.3	78.5	188.5	51.6	−1.5	0.14
BaDI	83.1	14.0	77.8	15.8	89	9.3	−1.7	0.08
mBI	95.8	8.8	99.3	1.8	92	11.7	−2.1	0.03
MoCA	25.7	3.2	26.8	2.1	24.5	3.8	−1.6	0.1
Hoehn and Yahr stage	-	-	-	-	3	0.9	n.a.	n.a.
UPDRS total score	-	-	-	-	33	14	n.a.	n.a.

Legend: F: female; M: male; WHS: Walking Handicap Scale; MTUAS: Media and Technology Usage and Attitudes Scale; BaDI: Barthel Dyspnea Index; mBI: modified Barthel Index; MoCA: Montreal Cognitive Assessment; n.a.: not applicable; UPRDS: Unified Parkinson’s Disease Rating Scale.

**Table 4 sensors-23-07238-t004:** Results of total Adherence-Days and Adherence-Repetitions considering all exercises together, and same values considering separately the patients’ performances related to motor and respiratory exercises.

	Total			COV19	pwPD		COV19 vs. pwPD COMPARISON	
	Mean	Std. Dev.	Mean	Std. Dev.	Mean	Std. Dev	Z	*p*
Adherence-Days Total	82.2	17.7	83.7	14.7	80.4	21.3	−0.07	0.9
Adherence-Reps Total	75.0	20.9	77.1	19.3	72.8	23.5	−27	0.79
Adherence-Days ME	83.1	18.5	84.5	16.2	81.5	21.5	−0.32	0.75
Adherence-Days RE	81.7	16.8	83.1	12.4	80.1	21.3	−0.23	0.80
Adherence-Reps ME	74.7	21.7	77.1	20.1	71.9	24.1	−0.18	0.86
Adherence-Reps RE	77.7	18.1	77.5	16.2	77.8	21.0	−0.04	0.97

Legend: Reps: repetitions; COV19: subjects with post-COVID-19 syndrome; pwPD: people with Parkinson’s disease; ME: motor exercises; RE: respiratory exercises.

**Table 5 sensors-23-07238-t005:** Results of the descriptive and pre–post-treatment comparative statistics.

	T0		T1		Pre–Post-Treatment Wilcoxon Signed Rank Test Results
	Mean	Std. Dev.	Mean	Std. Dev.	Z-Value; *p*-Value
BaDI WS	83.14	14.03	92.48	16.36	**−3.0; 0.003**
BaDI COV19	77.82	15.85	88.91	22.22	**−2.1; 0.04**
BaDI pwPD	89	9.26	96.4	3.69	**−2.4; 0.02**
mBI WS	95.81	8.77	97.19	6.65	−1.7; 0.09
mBI COV19	99.27	1.85	100	0	−1.3;.18
mBI pwPD	92	11.67	94.1	8.83	−1.2; 0.25
BFI WS	4.75	2.33	2.86	2.26	**−3.5; 0.0005**
BFI COV19	4.79	2.86	2.53	2.72	**−2.4; 0.01**
BFI pwPD	4.71	1.73	3.22	1.67	**−2.2; 0.03**
HR pre 2MWT, WS	88.48	11.95	85.29	15.94	−0.88; 0.38
HR pre 2MWT, COV19	91	10.17	85.64	13.79	−0.89; 0.38
HR pre 2MWT, pwPD	85.7	13.64	84.9	18.8	−0.10; 0.92
SpO^2^ pre 2MWT, WS	96.43	2.16	96.91	1.48	−0.59; 0.55
SpO^2^ pre 2MWT, COV19	96.46	2.42	97.18	1.47	−0.42; 0.67
SpO^2^ pre 2MWT, pwPD	96.4	1.96	96.6	1.51	−0.42; 0.67
HR post 2MWT, WS	96.71	15.13	101.38	14.09	−0.86; 0.39
HR post 2MWT, COV19	99.64	10.81	104.55	14.95	−0.40; 0.68
HR post 2MWT, pwPD	93.5	18.9	97.9	12.92	−0.89; 0.37
SpO^2^ post 2MWT, WS	96.38	1.56	96	1.70	−0.80; 0.42
SpO^2^ post 2MWT, COV19	97	1.61	96.36	1.29	−0.98; 0.33
SpO^2^ post 2MWT pwPD	95.7	1.25	95.6	2.07	−0.07; 0.94
2MWT (m) WS	128.95	29.52	143.24	30.31	**−3.3; 0.001**
2MWT (m) COV19	140.46	25.23	160.46	22.3	**−2.7; 0.005**
2MWT (m) pwPD	116.3	29.81	124.3	27	−0.1.8; 0.07
BDI WS	12.38	9.28	10.14	8.93	**−2.6; 0.01**
BDI COV19	11.91	9.17	9	9.61	**−2.5; 0.01**
BDI pwPD	12.9	9.87	11.4	8.44	−1.2; 0.23
BAI WS	13.19	9.62	9.62	11.93	**−3.1; 0.002**
BAI COV19	13.91	12.99	11.46	16.34	−1.6; 0.09
BAI pwPD	12.4	4.09	7.6	3.31	**−2.8; 0.005**
EQ-5D WS	8.381	1.884	7.524	1.83	**−2.6; 0.008**
EQ-5D COV19	8.455	2.067	7.455	2.21	−2.0; 0.05
EQ-5D pwPD	8.3	1.77	7.6	1.43	−1.8; 0.07
EQ-5D VAS WS	54.76	22.16	67.62	18.95	**−2.7; 0.007**
EQ-5D VAS COV19	62.73	17.08	68.18	22.72	−1.5; 0.12
EQ-5D VAS pwPD	46	24.59	67	14.94	−2.2; 0.03

Legend: WS = whole sample (N 21); COV19: subjects with post-COVID-19 syndrome; pwPD: people with Parkinson’s disease; HR = heart rate; 2MWT = 2-Minute Walking Test; m = meters; BaDI = Barthel Dyspnea Index; mBI = modified Barthel Index; BFI = Brief Fatigue Inventory; BAI = Beck Anxiety Inventory; BDI = Beck Depression Inventory; EQ-5D: Euro-Quality of life Questionnaire self-assessment-5 Dimension; VAS: Visual Analogic Scale. Statistically significant results are in bold.

## Data Availability

The data associated with the paper are not publicly available but are available from the corresponding author on reasonable request.

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
