# Peer review of "Telerehabilitation with ARC Intellicare to Cope with Motor and Respiratory Disabilities: Results about the Process, Usability, and Clinical Effect of the “Ricominciare” Pilot Study"

_sensors, 2023, doi:10.3390/s23167238_

Round 1

Reviewer 1 Report

This paper describes the implementation of a system that uses off-the-shelf MetaMotionR sensors combined with proprietary software. The paper is generally interesting to those who are measuring similar movement of the body using IMU sensors. The compliance measurements are beneficial to those looking to implement their proof-of-concept systems in a home environment.

There are several points that the authors need to clarify before the paper is in a publishable condition. These are:

The authors continually discuss their ARC system throughout this paper, but it is difficult to understand this system without visualising it and how the MetaMotionR sensors are located on the body. Some more information is therefore needed from the authors to help explain what is unique about this system when compared to other off-the-shelf systems. Therefore, I strongly recommend that the authors should take a photo of the system when the sensors are attached to the body, and a screenshot of the software would help the readers to visualise this system. And the authors should describe generally why this system is novel when using off-the-shelf MetaMotionR sensors.

A discussion on the accuracy and reliability of the MetaMotionR sensors would benefit the reader. The readers need to understand what accuracy should be expected from the sensors used in the ARC system. For example, the work by Kristen et al. (https://doi.org/10.1016/j.jbiomech.2019.109356) found that sensor placement affected accuracy of the MetaMotionR, and the type of movements performed by the wearer also influenced accuracy. Therefore, I recommend that the authors need to add a section in the paper to describe what levels of accuracy they should expect to obtain from the MetaMotionR sensors, and what error level is acceptable. This has not been discussed throughout the paper, but acceptable and expected levels of accuracy are fundamental to any type of measurement system and need to be discussed in detail.

Results in Table 2 show that heart rate and Spo2 measurements were captured before and after the 2MWT. Are these sensors part of the ARC system? Was this data captured at home by the users, or by a professional? The authors should clarify in a relevant section within the paper how these measures were captured (what sensors were used), and who collected this data.

The authors discuss health state in Table 2, and it is displayed ina box plot in Figure 5. How was health state calculated? The authors should explain how they derived health state and what measures were used to calculate it.

General

Some examples of repeated defining of acronyms such as PD which is defined in page 2, line 75, and then in page 4, line 159. And then the full name is provided in page 4, line 146.

I don’t understand what “POR FESR 2014-2020-Asse I COVID19)” refers to on page 4, line 153. I assume this is related to the MAGIC grant? Please clarify what this refers to.

The use of acronyms need to be re-examined as their use is not consistent throughout the paper.

Reviewer 2 Report

This study explores usability and feasibility of integrating a telerehab approach, based on the ARC intellicare system, to offer rehab therapy to people with respiratory issues du to COVID-19 and Parkinson’s disease. Though I fully support the importance of evaluating any technology on a usability and integration potential point of view, I have many concerns about this specific study and paper I would truly like the authors to address.

Main concern: Statistical analysis - There seems to be confusion in the different statistical approach and reporting or discrepancies between what was announced and what was performed (may also be a lack of details to ease comprehension). Few examples:

o   Section 2.4: inter-rater reliability was apparently verified and found to be -0.8, as assessed using a Cronbach alpha. However, a negative value usually means that something went wrong in the computation, either at data level or computation level itself… 

o   Section statistical analysis: the authors mention that mean and standard deviation will be used to describe continuous parametric variables. However, the notion of parametric / non-parametric refers mainly to the type of test that can be performed. Trying to understand this statement otherwise, I came to the potential explanation that normality was verified and if met, mean/stdev was used to describe? Is that the case?

o   Section statistical analysis: The authors announce, for example, that a Wilcoxon rank test will be used to assess pre-post changes, and that Mann-Whitney U test will be applied to compare between groups. However, results provide Z-scores… 

o   Section results: MTUAS score is reported to be “slightly related to age”, as assessed with a Spearman correlation. Yet, again Z score is given. Furthermore, a p-value of 0.48 is far from “slightly related”… Relation between MTUAS and MoCA is also reported using a Z score.

o   Section usability and acceptability: The authors report that COV19 SUS scores varied from 80 to 75, while PD SUS scores evolved from 74 to 80. Then, it is mentioned that no statistically significant differences were found between groups. Was that analysis performed at baseline, follow-up or on change?

o   Table 1: Z-scores are reported while Mann-Whitney U test was announced… Which one was truly performed?

o   Table 2: Wilcoxon signed rank tests results giving a Z value?

Other concerns / issues:

-       Very limited information was given on the system used. Indeed, apart saying that it uses AI with 5 IMU sensors, no other information is available… Specifically:

o   How are the sensors apposed? On which segments? How affixed? 

o   Was the AI algorithm validated? On which population? 

o   Adherence is assessed with the number of reps performed… was that measure validated? Even for a population that may have problems performing the movement?

o   Problems appear to be registered only as “adverse events”… but does the system give a feedback on the quality of movement as the person would get in an “in-person” rehab session? Risk of injuries?

o   How was support provided if required?

-       Limited info on the protocol:

o   We know that 5 persons were involved in the protocol (MGC, MC, CR, FAB first announced… then talk about MEL and CM)… Yet, was the rehab protocol personalized? Who was blind? Was the same therapist involved at all time points with the patients?

o   It appears participants were excluded for not having a caregiver that could potentially assist them. Yet, this was not exposed in the inclusion/exclusion criteria and no information on the reason for this is given in the paper.

o   Was the training on the system (technology) or the rehab program itself?

-       Results that would be interesting or that are currently announced but appear to be missing:

o   Any technological problems encountered during the study? (Live or off-line)

o   Any adverse events?

o   Results include “Health State”… Please clarify how this measure was achieved? Specific scale (not found in method table).

-       Discussion: At this stage, it is difficult to judge upon the discussion as I question the analysis. Yet, few questions arose:

o   SUS scores globally remained stable, but PD score improved. Discussion focusses on this point, describing that support/structure may have helped. However, the fact that COVID patients score decreased was not discussed…

o   No statistical improvement in 2MWT in PD population was discussed based on the assumption that walking test scores may be more dependent on medication than cardio-fitness level for moderate-advanced PD. However, no description of PD patients was given (Hoehn & Yarh? UPDRS?). Furthermore, last sentence does not seem to be supported by any results from this study (especially with the limited sample size). 

Other comments:

-       Please review title to announce that this is pilot study results… 

Long sentences shall be reviewed.

Repetitions to avoid (e.g. thanks to IMU... 3x)

Round 2

Reviewer 2 Report

Thank you for the considerations given to my previous comments.

I would recommend an overall revision to improve further the manuscript.